# Sentinel-2 Data for Land Use Mapping: Comparing Different Supervised Classifications in Semi-Arid Areas

Khouloud Abida [1],*, Meriem Barbouchi [2], Khaoula Boudabbous [1], Wael Toukabri [2], Karem Saad [3], Habib Bousnina [1] and Thouraya Sahli Chahed [4]

[1] National Institute of Agronomy of Tunisia (INAT), Carthage University, Avenue Charles Nicolle, Tunis 1082, Tunisia
[2] Laboratoire Sciences et Techniques Agronomiques (LR16INRAT05), National Institute of Agricultural Research of Tunisia (INRAT), Carthage University, Tunis 1004, Tunisia
[3] Ecole National des Ingénieurs de Sfax (ENIS), Carthage University, Sfax 3038, Tunisia
[4] National Centre for Mapping and Remote Sensing, Ministry of National Defense (CNCT), Tunis 1080, Tunisia
* Correspondence: abidakhouloud22@gmail.com; Tel.:+216-22-676-333

**Abstract:** Mapping and monitoring land use (LU) changes is one of the most effective ways to understand and manage land transformation. The main objectives of this study were to classify LU using supervised classification methods and to assess the effectiveness of various machine learning methods. The current investigation was conducted in the Nord-Est area of Tunisia, and an optical satellite image covering the study area was acquired from Sentinel-2. For LU mapping, we tested three machine learning models algorithms: Random Forest (RF), K-Dimensional Trees K-Nearest Neighbors (KDTree-KNN) and Minimum Distance Classification (MDC). According to our research, the RF classification provided a better result than other classification models. RF classification exhibited the best values of overall accuracy, kappa, recall, precision and RMSE, with 99.54%, 0.98%, 0.98%, 0.98% and 0.23%, respectively. However, low precision was observed for the MDC method (RMSE = 1.15). The results were more intriguing since they highlighted the value of the bare soil index as a covariate for LU mapping. Our results suggest that Sentinel-2 combined with RF classification is efficient for creating a LU map.

**Keywords:** sentinel-2; land use mapping; supervised classification; spectral index; machine learning

## 1. Introduction

Land use (LU) is the result of complex interactions between humans and the physical environment, having significant effects on ecosystem processes [1,2]. Furthermore, it is strongly associated with the sustainable development of the social economy [3]. Land uses change more quickly when economic development picks up speed and the contrast between different LU types increases [4]. In fact, the surface of the earth is rapidly changing due to certain natural reasons and other impacts by society. Thus, various approaches were used to detect and map LU [5].

In fact, there is a high demand for LU maps for the monitoring and management of the most significant changes in the environment, including urbanization, agricultural expansion and the creation of strategies to understand soil biophysical processes [6,7]. Therefore, LU maps are indispensable for controlling the dynamics of environmental ecosystems and to monitor the environmental phenomena [8,9]. Sustainability science is based on LU mapping [10]. It provides a better distinction between artificial areas, agricultural areas, forests, moors and wetlands [11] and monitors the direct effect of climate change, as well as changes in vegetation cover, surface conditions and LU on catchments [12].

Both the management of land potential throughout the year and the monitoring of changes in the environment can benefit from accurate LU maps. Various tools can be used to obtain LU mapping. One of the primary techniques that can be applied to achieve land

cover (LC) mapping is remote sensing [7,10,13,14]. According to [11,15], the contribution of remote sensing in conventional mapping methods is indispensable to making cover mapping a reasonable practice.

Optical remote sensing images are considered a very appropriate tool for studying and managing the evolution of land use and land cover (LULC) in a constant and continuous way [16,17]. Abou Samra et al. [18] focused on the potential benefits of optical image data in order to assess human-induced changes and their impacts and to highlight how remote sensing contributes to developing the best environmental management plans for mid-crop climate and land use change. Furthermore, in arid and semi-arid areas, Shafizadeh-Moghadam et al. [19] revealed that multispectral data are an effective tool for LULC mapping, especially for the separation of different green cover areas. Optical remote sensing data from Sentinel-2 has been increasingly successful due to its high spatial resolution and improved spectral resolution in the near-infrared region, which provides wide applicability for classification and calculating soil properties [20], land mapping [21] and monitoring purposes [15,22,23].

LU classification is one of the most popular applications used in remote sensing [16]. Using the pixel values, different techniques are adopted for data extraction of LC types from optical imagery (Sentinel-2). In this regard, the classification method is the most useful tool for image interpretation and information extraction in different bands of the satellite sensor. This information is extracted in terms of digital numbers that will be converted to a category [24].

In recent years, a great revolution in the implementation of various classification algorithms has been achieved. Classification can be carried out by different methods, parametric or nonparametric, contextual or noncontextual [25] and supervised or unsupervised methods [26].

Supervised classification is considered a technique where the user supervises the process of classifying the pixels. The user specifies the various pixel values or spectral signatures that might be associated with the specific class [24]. Through supervised classification, information such as LC type, vegetation type and soil properties can be obtained [27]. Moreover, referring to [27,28], supervised classification is more advantageous than unsupervised classification in most applications [29]. It has been commonly employed to detect the land use map [19], and according to [30], it must be followed by knowledge-based specialist classification systems depending on reference maps to enhance the accuracy of the classification process.

Machine learning algorithms are widely used in remote sensing data [10,14] and applied for LU classification [28,31]. Performance criteria are optimized by using data to program computers in machine learning [32]. The latter has a wide range of uses and is one of the fields with the fastest growth rate. In addition, one of the targets of the machine learning technique is to give the algorithm the ability to learn, implement [29] and improve the efficiency of systems and the de-signature of machines [33].

Recent recommendations for classifying land cover include different methods, such as Random Forest. By comparing the accuracy of RF algorithms with other models, such as support vector machine (SVM) and artificial neural network (ANN), for LU mapping using optical image data, Talukdar et al. [34] have shown the effectiveness of using RF and revealed that all the classifiers have a similar accuracy level with minor variation, but the RF algorithm has the highest accuracy.

Additionally, it was discovered by Kulkarni and Lowe [35] that RF outperformed all other classifiers, including MDC, in terms of total accuracy and kappa coefficient. Added to that, Shareef et al. [36] have shown that when RF and KDTree-KNN classification are compared for the purpose of creating LULC maps using multispectral images, RF always has higher accuracy than KDTree-KNN.

Despite the previous studies using the RF algorithm in many data mining applications, its potential is not fully explored for analyzing Sentinel-2 images to create LU maps. In recent decades, Tunisia, with a semi-arid climate and irregular rainfall, has been threatened

by an increase in soil degradation associated with a change in land usage [33,34]. In this context, the main objectives of this study are to (i) compare the performance of different supervised classification algorithms of LU mapping using Sentinel-2 image data and (ii) assess the most efficient supervised classification for mapping LU and the potential of Sentinel-2 data to classify LU in a semi-arid area.

## 2. Materials and Methods

### 2.1. Study Area

The study area is located in Tunisia's Northeast, Zaghouan governorate (latitude: 36°32′47.81″ N, longitude: 10°2′7.46″ E) (Figure 1C), with a total area of 34,000 ha. The topographic elevation in the research region ranges from 130 to 1295 m in the output of the Meliane valley's watershed and at the top of Zaghouan mountain, respectively [37]. The region is characterized by a semi-arid climate with two seasons: a dry season from June to September and a wet season from October to May. Rainfall varies greatly in both time and place. The annual evapotranspiration rate is 1398 mm, while the average rainfall is 390 mm. The average temperature value is 22 °C [38]. The study area is composed of vast plains with a series of mountainous masses in the middle, crossed in its south-eastern part by the limestone Dorsale, which has played an important paleogeographic and tectonic role in geological eras [39]. Moreover, this plain is a collapsed area limited from the north by the Triassic fault of the mountain [37].

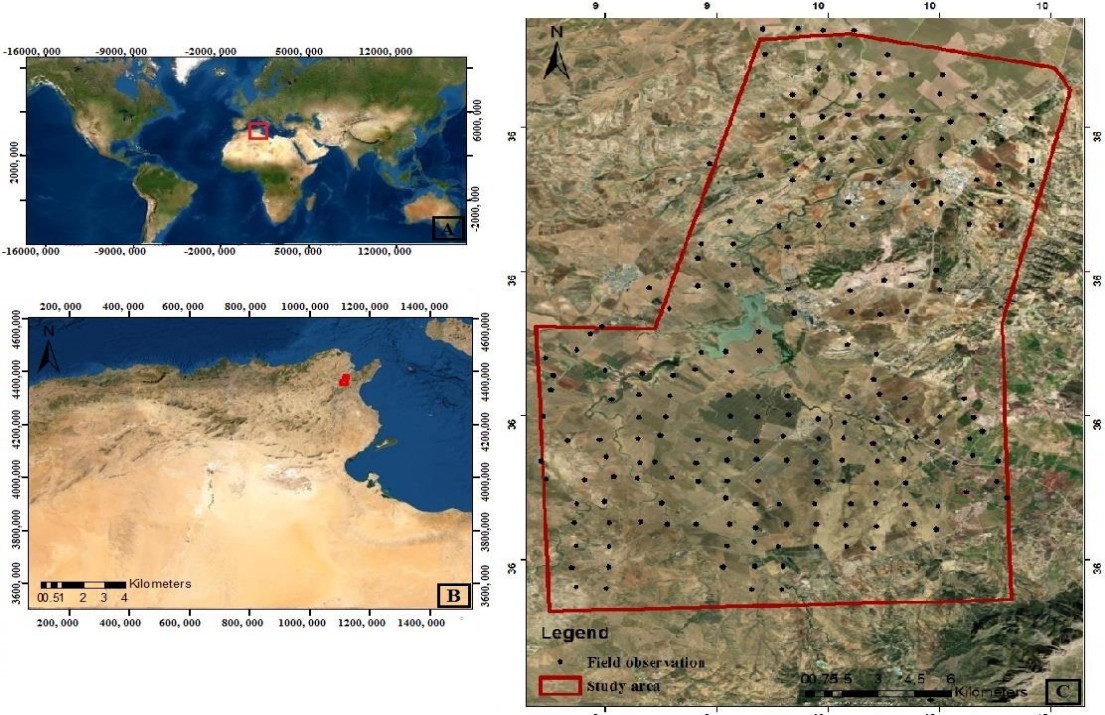

**Figure 1.** Location of the study area. (**A**) Tunisia's location worldwide, (**B**) Governorate of Zaghouan in Tunisia and (**C**) Study area.

Further, the vegetation is quite homogenous and characterized by shrubby species such as (*Calycotome villosa*) and clumps of esparto grass (*Stipa tenacissima* L.) with some spontaneous annual crops. Added to that, fruit growing with intercropping of irrigated vegetable crops and field crops are installed [40,41].

### 2.2. Data Collection

The dataset used in this study was obtained from the regional agricultural development committee of Zaghouan (RADCZ of Tunisia in 2021). It consists of LU data, boundaries of the study area, topography and climatic data [42].

Data collection took place at the end of August month. The chosen point and species crops are adopted based on data obtained from Google Earth (Table 1), Tunisian LU maps and, finally, by a field survey (Figure 1).

**Table 1.** Data collection information.

| Data Collection | Characteristics |
|---|---|
| Sample number | 220 points |
| Date | August 2021 |
| Sampling type | systematic with a step of 1 Km |
| Total area | 34,000 ha |
| Sol class number | 6 classes |
| Water | 4 points |
| Urban area | 42 points |
| Fields crop | 67 points |
| Arboriculture | 65 points |
| Forest | 34 points |
| Bare soil | 8 points |

In addition to the field data, in our study, we used Sentinel-2 images, which have 13 spectral bands with different spatial resolutions: 10, 20 and 60 m [43].

For LU classification, the Sentinel 2-MSIL 1C images were acquired from ESA's Copernicus program [44]. Optical data were extracted in August 2021 using the Sentinel-2 satellite with a high spatial resolution (HR) with cloud cover not exceeding 1.88% and covering the entire study area. SNAP 8.0 software (European Space Agency) was used to pre-process the S2 images, including radiometric calibration and atmospheric correction.

*2.3. Methodology*

The overall workflow of the study is described in Figure 2. This methodology is divided into four steps: pre-processing, processing, supervised classification and classification's accuracy evaluation.

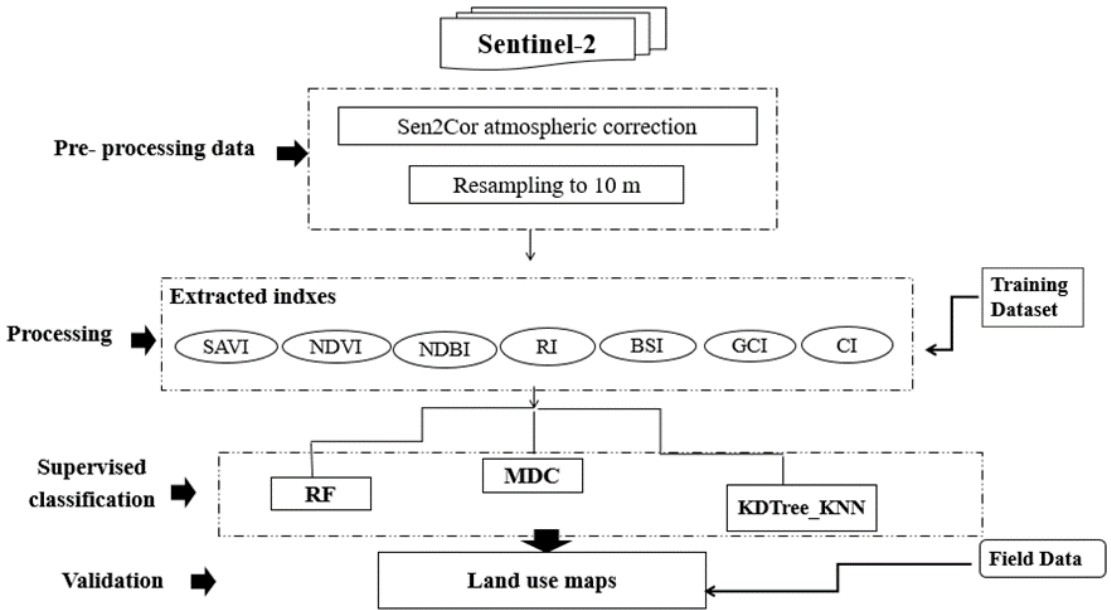

**Figure 2.** Workflow of land use classification with Sentinel-2 imagery.

### 2.3.1. Pre-Processing

Pre-processing was carried out using the SNAP tool provided by ESA. First, the image was resampled to 10 m; secondly, to avoid distortion that may occur during the data collection, Sens2Cor atmospheric correction was performed [22] before information extraction and, finally, the study area was subset.

### 2.3.2. Processing

In the processing step, six classes were identified (urban area, water, forest, field crops, arboriculture and bare soil). After pre-processing the S2 image, the training region polygons for each class were created. In this research, the input layers used in the model prediction consisted of five bands: B2, B3, B4, B8 with 10 m resolution and B11 because we need it in the calculation of the bare soil index. In addition to these spectral bands, vegetation and soil conditions of the study area were defined by the Normalized Difference Vegetation Index (NDVI), Coloration Index (CI), Redness Index (RI), Normalized difference built-up index (NDBI), Adjusted Vegetation Index (SAVI), Green Leaf Index (GLI), Green Chlorophyll Index (GCI) and Bare Soil index (BSI). All these indices were used to increase the classification accuracy (Table 2).

**Table 2.** Descriptions of some characteristic indices.

| Index | Characteristics | Equations | Reference |
|-------|-----------------|-----------|-----------|
| NDVI | Normalized Difference Vegetation Index: is used to monitor the condition of vegetation cover as well as to evaluate the photosynthetic activity of vegetation. Water, bare soil and vegetation are indicated by NDVI values of $-1$, 1 and 0, respectively. | $(B4 - B8)/(B4 + B8)$ | [45,46] |
| CI | Coloration Index: Applying this index, researchers can learn more about the soil's organic matter content and mineral composition. | $(B4 - B3)/(B4 + B3)$ | [45,47] |
| RI | Red Index: Used as one of the indices to assess soil mineralogy, including iron concentration. | $B4^2/B3^2$ | [48] |
| BSI | Bare Soil Index: Provides an idea of the state of crops, and allows the detection of recent deforestation or monitoring of droughts. Used to improve the accuracy of bare soil prediction using medium-resolution satellite data. | $[(B11 + B4) - (B8 + B2)]/[(B11 + B4) + (B8 + B2]$ | [49–52] |
| GCI | Green Chlorophyll Index: This index is used to calculate the amount of leaf chlorophyll in a wide variety of plant species. It decreases in stressed plants, making it a useful indicator of plant health. | $[B8/B3] - 1$ | [53–55] |
| SAVI | The Soil-Adjusted Vegetation Index: The ground-adjusted vegetation index was designed to minimize the influences of soil brightness. Its creator, Huete, added a soil adjustment factor L to the NDVI equation to correct for the effects of soil noise (soil color, soil moisture, soil variability across regions, etc.), which tend to have an impact on the results: Important fact: L is a soil brightness correction factor ranging from 0 to 1. | $(B8 - B4)/(B8 + B4 + L) \times (1 + L)$ Avec L = 0.428 | [56] |
| NDBI | The normalized difference built-up index: To enhance the built-up area's ability to predict future incidents using medium resolution satellite datasets | $(B8 - B4)/(B8 + B4)$ | [49–52] |

### 2.3.3. Supervised Classification Techniques

Three popular classifiers were applied in this study, Random Forest (RF), the Minimum Distance Classification (MDC) and the K-Dimensional Tree-KNN classifications technique, and were implemented in SNAP. The supervised classification consists of three steps;

selecting the training area, creating signature fields and classifying the S2 image [24]. In a specific limited region, different combinations of feature sets were evaluated.

Machine Learning Algorithms

Based on previous studies, this section provides descriptions of the advantages of the three algorithms used in the supervised classification.

Random Forest Classification

The RF classification has been successfully used to integrate remotely sensed imagery with ancillary geographic information for LULC and is the most popular among digital soil mappers for predicting soil properties. The predictive performance of RF is better than other ML techniques for soil mapping [57–60]. A deep mathematical description of RF is acquired by [61].

Minimum Distance Classification

Minu Nair and Bindhu [27] have shown that the MDC depends mainly on the training dataset and is recognized to have rapid execution with all pixels being well classified.

K-Dimensional Tress-k-Nearest Neighbor

The KDTree-KNN model can greatly improve search performance while reducing time complexity. It is characterized by a low cost and effort for the learning processes and no advanced design and training are required [27,62].

2.3.4. Validation

To validate the results, we used the confusion matrix, and from this, several external measures were defined, namely:

The Accuracy, which gives an overall indication of the matching degree between the model and the ground truth. It represents the ratio of the sum of well-classified pixels to the sum of classified pixels

$$\text{The Accuracy} = \frac{true\ positive + true\ negative}{true\ positive + false\ positive + false\ negative + true\ negative} \tag{1}$$

The Precision index, which gives an idea about the correct prediction rate of positive values and recall; it represents the true positive rate or the sensitivity. Precision and recall are defined as:

$$\text{Precision} = \frac{\text{True positive}}{\text{True positive} + \text{False positive}} \tag{2}$$

$$Recall = \frac{True\ positive}{True\ positive + False\ negative} \tag{3}$$

In addition, the Kappa coefficient is calculated by the following formula [63]:

$$\text{Coefficient } Kappa = \frac{N \sum x_{ii} - \sum x_{i+}\ x_{+i}}{N^2 - \sum x_{i+} x_{+i}} \tag{4}$$

where N: number of rows and columns in the confusion matrix, $X_{ii}$: observation in row i and column i, $X_{i+}$: Marginal total of row i, $X_{+i}$: marginal total of column i.

## 3. Results

### 3.1. Statistical Evaluation

The statistical results are presented in Table 3 the obtained values in this study were run based on the results classification. These detailed values were obtained in terms of the fully supervised creation of the polygon area. The latter was evaluated with Google Earth.

**Table 3.** Statistical result analysis.

| Sentinel2 31 August 2021 | Accuracy Value (%) | RMSE | Kappa Value (%) | Precision (%) | Recall (%) |
|---|---|---|---|---|---|
| RF | 99.54 | 0.23 | 0.98 | 0.986 | 0.986 |
| MDC | 94.29 | 1.15 | 0.79 | 0.828 | 0.83 |
| KDTree-KNN | 99.07 | 0.49 | 0.96 | 0.972 | 0.97 |

In our investigation, the data revealed that the obtained accuracy values were more than 94%, and the highest value occurred for the RF classification, 99% (Table 3). In regards to the Kappa, RMSE, Precision and Recall values, the best values were noted for the RF classifications, while we exhibited the worst RMSE value for the MDC classification algorithm, with 1.15.

The description of the Kappa value can be observed in Table 3.

### 3.2. Land Use Map

Among the three obtained LU maps, our data revealed that the six classes of the LU map depended on the choice of the supervised algorithm. In fact, according to the LU field survey and Google Earth, the RF classification, with homogeneity of class distribution (water, forest, urban area, bare soil, fields crop and arboriculture), is more realistic (Figure 3). Moreover, MDC and KDTree-KNN provided a big difference in the obtained maps, where the field crop class is the most dominant in the produced map, showing, in most cases, a lack of precision.

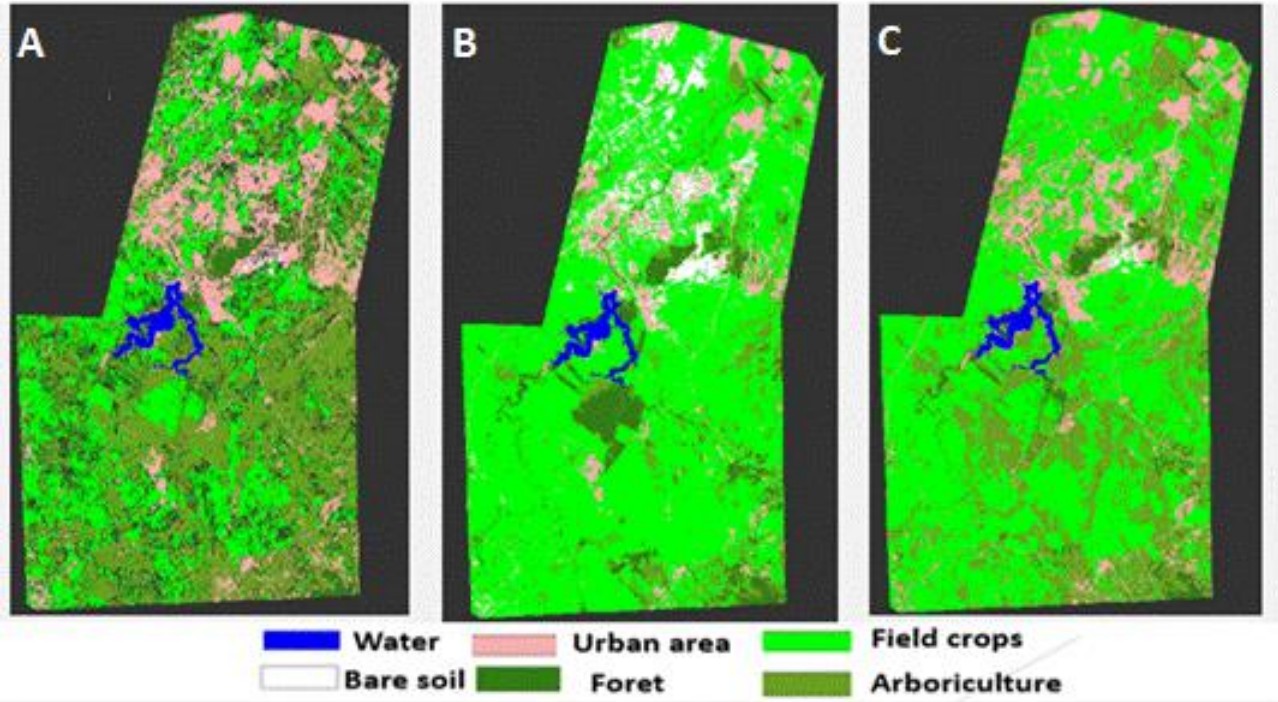

**Figure 3.** Land use maps for the three classifications. (**A**): Random Forest classification, (**B**): MD classification and (**C**): KDTree-KNN classification.

RF showed the best class distribution; this is consistent with the statistical results obtained by the confusion matrix. However, in the two other models (MDC and KDTree-KNN), there is some ambiguity between the percentages of class distribution and the land use map obtained (Figure 4).

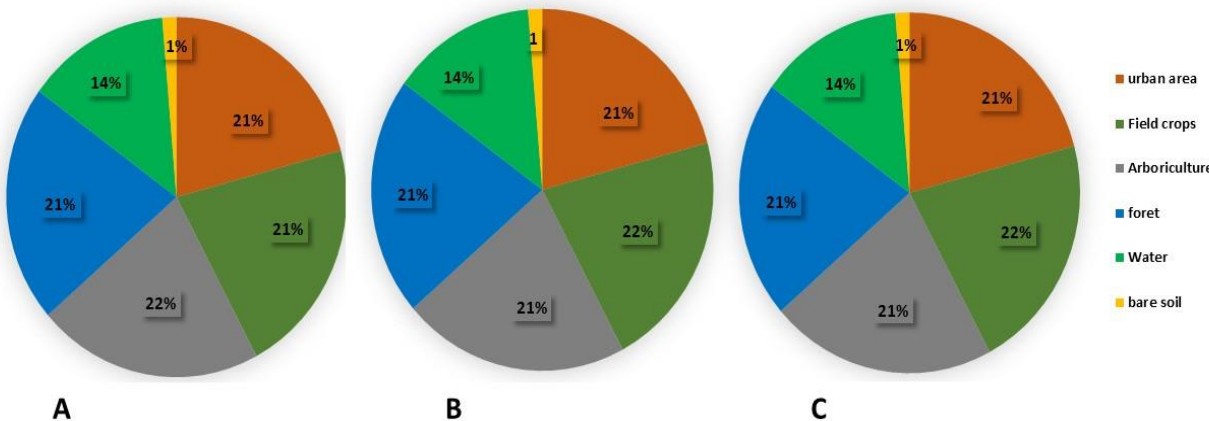

**Figure 4.** Results for class distribution for the three algorithms ((**A**); RF, (**B**); MDC and (**C**); KDTrees-KNN).

### 3.3. Relative Importance of the Covariates in RF and Class Distribution

In our investigation, the correlation of the index distribution (GCI, NDVI, RI, NDBI, SAVI, CI and BSI) in the RF classification used in this investigation showed that the highest value was obtained for the Bare Soil Index (BSI), with more than 25% (Figure 5), and the lowest correlation values were obtained for NDVI and GCI (less than 7%) (Figure 5).

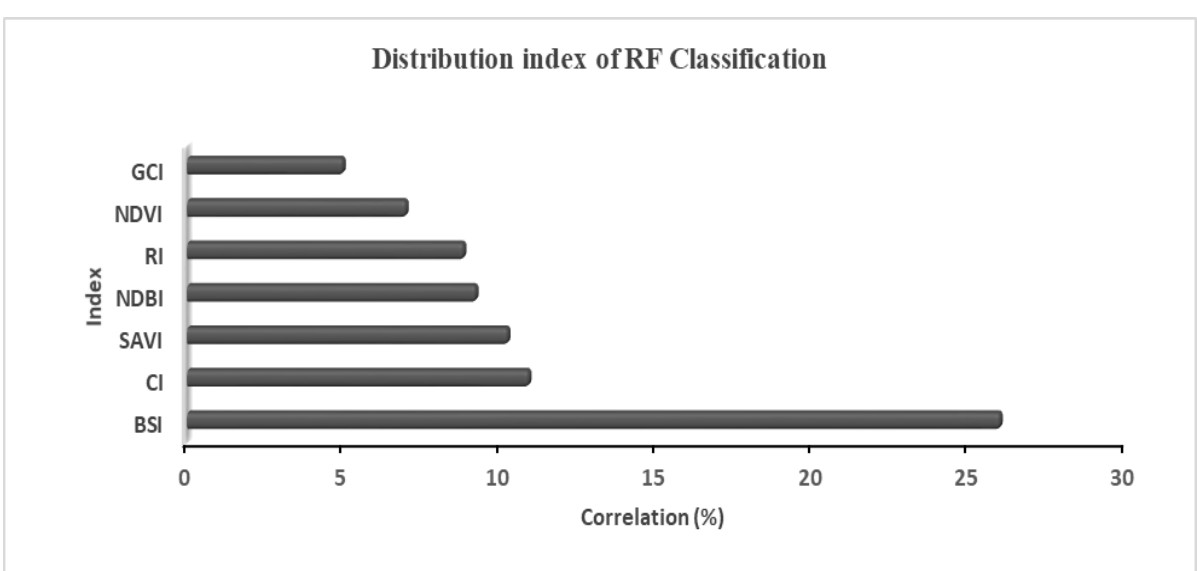

**Figure 5.** Percentage of index correlation for RF classification. GCI: Green Chlorophyll Index; * NDVI Normalized Difference Vegetation Index,* CI: Coloration Index;* RI: Redness Index;* NDBI: Normalized difference built-up index;* SAVI: Soil Adjusted Vegetation Index;* BSI: Barren Soil index.

### 4. Discussion

In recent years, with the effect of climate change, the application of optical remote sensing has been used to monitor and assess the distribution of land area. Many tools have been investigated, but the use of Sentinel-2 images with machine learning algorithms has been less studied, particularly in Tunisia's semi-arid climate context.

The present research addressed the research questions: (1) How can Sentinel-2 images and the use of machine learning algorithms be efficient for managing soil use? (2) How do the three supervised classification algorithms for the creation of the LU map compare? (3) What are the best performance spectral indices of the supervised classification algorithms for producing a LU map?

### 4.1. Comparison between Machine Learning Classification Methods

The results show that the best machine learning algorithm was obtained for RF classification. It performed better in terms of Kappa value, RMSE, Precision, Recall and Accuracy of LU classification. The LU visualization obtained by RF classification is the most efficient and close to reality. Our result revealed that RF classification is more efficient than MDC and KDTree-KNN and has a better classification accuracy (99.54%); this is in agreement with the studies [35,36], and such results could be explained by the fact that it is simple and easy and takes less time [64,65].

More interestingly, the RF method is based on decision tree classifiers. Therefore, it is more flexible and useful for classifying optical image data than MDC. The RF methods used to enhance the predictive accuracy for LU classification subsequently help to control the overfishing of data [66]. As demonstrated by [67], RF was able to successfully separate field crops and arboriculture from the other LU types for all settings.

In our study, the best accuracy (99.54%) obtained for LU classification using Sentinel-2 is in line with the results achieved by [6,68] in mapping the forest cover in Vietnam (in Southeast Asia) and in Gabon (in Central Africa), respectively. In the Mediterranean climate (in Northwestern Spain), Sentinel-2 data and RF supervised classification for an LC map [69] showed an overall accuracy of 91.6%, which is close to our obtained results.

### 4.2. Spectral Index Importance for Land Use Classification

To identify both the vegetation and soil characteristics for the LU area, a combination of various indices, such as the NDVI, SAVI, BSI, CI, RI, NDBI, GLI and GCI, was adopted for more precision. Our data revealed that BSI has the best correlation, more than 25%; this is in line with [66] using multispectral imagery. The BSI performance improves with increasing aridity and decreasing soil moisture. In addition, Qiu et al. [70,71] have shown that the percentage of bare soil often decreases when the LC is transferred from grass and crops to forests. Moreover, optical image data are a reliable approach for detecting bare soil due to seasonal bare soil [51]. This highlights the fact that BSI allows the differentiation between bare soil and built-up soil, which improves the evaluation of urban expansion and the good management of agricultural land [72]. In contrast to BSI, our investigation showed that the correlation between NDVI and GCI is less than 8%.

Several remote sensing indices are derived from different spectral wavelengths of the optical image, which aim to enhance and separate bare soil from other LC features (water, forests, urban areas . . . ) [73].

In our case, the enhancement of the BSI value has been promoted by the semi-arid climate, the low cloud cover percentage, and the field campaign period (August). In our study, the results proved that Sentinel-2 can be an efficient tool for describing soil characteristics using a spectral index. Such results corroborate those obtained in [74] in India, showing the preference of Sentinel-2 data over other multispectral data for the classification of vegetation types. Therefore, the decrease in the NDVI values is explained by the lower natural vegetation cover [66]. In the case of multispectral images, Naguyen et al. [51] and Rasul et al. [75] indicated that the study area's climatic conditions, surrounding vegetation, soil composition and moisture content must be taken into consideration in order to establish a satisfactory correlation for the indices.

### 5. Conclusions

The recent research emphasizes the relevance of Sentinel-2 images for managing LU and creating LU mapping using machine learning classification methods.

The highest accuracy, Kappa value, Recall and Precision for the LU classification were obtained for RF. The best correlation for the spectral index used for LU classification was observed in BSI, and the lowest was for GCI and NDVI. The significant importance of optical remote sensing images using supervised classification indicated their powerful capacity for mapping and LU management potential.

However, the research area's climatic conditions and soil qualities must be taken into account while choosing a classification algorithm. Future studies should incorporate hyperspectral and SAR images with very high resolutions for soil classification to monitor changes in LU and support agriculture in managing the land.

**Author Contributions:** Conceptualization. M.B; methodology. M.B. and K.A.; validation. H.B. and M.B. formal analysis. K.A., M.B. and H.B.; investigation. K.A., M.B., H.B. and K.B.; resources. T.S.C. and M.B., data curation. K.A., M.B. and K.B. writing—original draft preparation. K.A., M.B. and K.B., writing—review and editing. M.B., H.B., K.B., W.T. and K.S.; supervision. H.B.; project administration. T.S.C. and H.B.; funding acquisition. H.B. All authors have read and agreed to the published version of the manuscript.

**Funding:** This research was funded by the Centre for Cartography and Remote Sensing (NCCRS) in Tunisia, as part of a soil properties mapping (SPM2) project and Team of Geographic Information System of Soil Resources in Tunisia (GISSRT).

**Institutional Review Board Statement:** Not applicable.

**Informed Consent Statement:** Not applicable.

**Data Availability Statement:** Not applicable.

**Acknowledgments:** We are grateful for the support of the entire soil properties mapping (SPM2) project team at the National Mapping and Remote Sensing Centre (NMRSC) in Tunisia for their professionalism, efforts and their encouragement, all the staff at the Regional Agricultural Development Committee of Zaghouan (RADCZ) for their availability, benevolence and help. We are grateful to the team at the Tunisian Higher Institute of Agronomic Research (INRAT) and the National Agronomic Institute of Tunisia (INAT) for their guidance and collaboration.

**Conflicts of Interest:** The authors declare no conflict of interest.

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
