# Peer review of "Sentinel-2 Data for Land Use Mapping: Comparing Different Supervised Classifications in Semi-Arid Areas"

_agriculture, doi:10.3390/agriculture12091429_

Round 1

Reviewer 1 Report

Dear authors,

I have gone through the manuscript

 Sentinel-2 data for land use mapping: comparing different su-2 pervised classifications in semi-arid areas.. The author has done good work but still this manuscript needs more attention of the author.

1-   The  written English language need to be revised

2-   In line 47, page 1   surface conditions, and s LU on catchments please remove s

1-     Introduction section: There is no sufficient related literature review about using remote sensing to map LCLU changes and using Sentinel 2. Please add some specific literature. the author add only one line  in the introduction . According to [20] it is easier 59 to identify floods from optical images than from radar SAR images. I think the authors could read them and cite these. please see

Abou Samra R.M. (2022). Dynamics of human-induced lakes and their impact on land surface temperature in Toshka Depression, Western Desert, Egypt. Environmental Science and Pollution Research, 29, 20892-20905.

Abou Samra R.M. & Ali R. (2022). Monitoring of oil spill in the offshore zone of the Nile Delta using Sentinel data. Marine Pollution Bulletin, 179, 113718.

2-   Please describe the research gap in the introduction section

3-   Please add description of meteorological data of the study area.

4-   The author did not make much contribution to the field of study Please increase resolution of figure 4

5-   This write-up did not address the current limitations in this field of study.

6-    Please add grid to figure 1

7-   In line 256, page 12  using machine learning algorithm machine learning algorithms could be efficient, please remove repeated words

Author Response

First, we would like to thank both reviewers for their constructive and encouraging feedback which improved our manuscript (agriculture-1885143) considerably. We made all the revisions according to reviewer concerns. Please find the answers for each of the points raised by the reviewers in the present MS word file. Please refer to the responses as follows: Reviewers comments in normal font, author’s response in blue, changes in the MS in red.

Reviewer' responses

 Reviewer 1:

General comments

 Dear authors, I have gone through the manuscript Sentinel-2 data for land use mapping: comparing different supervised classifications in semi-arid areas. The author has done good work but still this manuscript needs more attention of the author.

Comment 1:   The written English language need to be revised.

              Reply 1: As recommended, the english language was revised, all manuscript was checked carefully.

Comment 2:  In line 47, page 1 surface conditions, and s LU on catchments please remove s

              Reply 2:  As requested, the “s” was removed. (L55).

Comment 3: Introduction section: There is no sufficient related literature review about using remote sensing to map LCLU changes and using Sentinel 2. Please add some specific literature. the author add only one line in the introduction. According to [20] it is easier 59 to identify floods from optical images than from radar SAR images. I think the authors could read them and cite these. please see

Abou Samra R.M. (2022). Dynamics of human-induced lakes and their impact on land surface temperature in Toshka Depression, Western Desert, Egypt. Environmental Science and Pollution Research, 29, 20892-20905.

Abou Samra R.M. & Ali R. (2022). Monitoring of oil spill in the offshore zone of the Nile Delta using Sentinel data. Marine Pollution Bulletin, 179, 113718.

                      Reply 3: Thank you for your comment, the fisrt introduction section was improved as recommended. As suggested, the two above references were added. We read carefully the the sentence “According to [20] it is easier 59 to identify floods from optical images than from radar SAR images”, and accordingly, it was removed.   (L56-59)

Comment 4:   Please describe the research gap in the introduction section.

                 Reply 4: We have taken into account your suggestion, the research gap in the introduction section was well improved. (L56-61, L72-84, L88-104).

We have added some new references to improve our manuscript:

  1. Abou Samra, R.M. Dynamics of human-induced lakes and their impact on land surface temperature in Toshka Depression, Western Desert, Egypt. Environmental Science and Pollution Research.2022, 29, 20892-20905. [https://doi.org/10.1007/s11356-021-17347-z]

19.Shafizadeh-Moghadam,H,; Khazaei,M,; Alavipanah,S.K,; Weng,Q. Google Earth Engine for large-scale land use and land cover mapping: an object-based classification approach using spectral, textural andtopographical factors. GIScience & Remote Sensing.2021, 58,:914–928 [https://doi.org/10.1080/15481603.2021.1947623].

25.Keuchel,J,; Naumann,S,; Heiler,M,;  Siegmund,A. Automatic land cover analysis for Tenerife by supervised classification using remotely sensed data. Remote Sensing of Environment.2003, 86: 530–541.[ doi:10.1016/S0034-4257(03)00130-5].

  1. Osisanwoer Supervised Machine Learning Algorithms: Classification and Comparison. International Journal of Com-puter Trends and Technology (IJCTT) .2017, 48, Number 3 June 2017:128-138.

        30 Sahebjalal,E,; Dashtekian,K. Analysis of land use-land covers changes using normalized difference vegetation index (NDVI) differencing and classification methods. African Journal of AgriculturalResearch.2013, 8(37): 4614-4622[DOI:10.5897/AJAR11.1825].

  1. Kotisiantis,S.B. Supervised Machine Learning:A review of classification techniques.2007,160:1-24

       34.Talukdar,S, ; Singha,P, ; Mahato,S,; Shahfahad,; Pal,S,; Liou,Y.A,; Rahman,A. Land-Use Land-Cover Classification by Machine Learning Classifiers for SatelliteObservations—A Review. Remote Sens. 2020, 12, 1135 [doi:10.3390/rs12071135).

  1. Kulkarni, A.D, ; Lowe,B. Random Forest Algorithm for Land Cover Classification. International Journal on Recent and Innovation Trends in Computing and Communication.2016,4: 58-63.

      36.Shareef, M. A,; Hassan, N. D,; Hasan, S. F,; Khenchaf, A. Integration of Sentinel-1A and Sentinel-2B Data for Land Use and Land Cover Mapping of the Kirkuk Governorate, Iraq. International Journal of Geoinformatics.2020,16 :87-96.

      37.Ameur,M,; Hamzaoui-Azaza,F, ; Gueddari,M. Suitability for human consumption and agriculture purposes of Sminja aqui-fer groundwater in Zaghouan (north-east of Tunisia) using GIS and geochemistry techniques. Environ Geochem Health .2016,38:1147–1167.[ DOI 10.1007/s10653-015-9780-2].

  1. Mejri,S,; Chekirbene, A,; Tsujimura ,M,; Boughdiri, M,; Mlayah, A. Tracing groundwater salinization processes in an inland aquifer: A hydrogeochemical and isotopic approach in Sminja aquifer (Zaghouan, northeast of Tunisia). Journal of African Earth Sciences.2017:1-39. [10.1016/j.jafrearsci.2018.07.009].

Comment 5:   Please add description of meteorological data of the study area.

                 Reply 5: Your point of view has been considered, the description of meteorological data of the study area was added asThe region is characterized by a semi-arid climate with a rainy season from October to May and a dry season from June to September. Rainfall is highly variable in space and time. the Average rainfall is 390 mm and evapotranspiration is 1398 mm/year.The annual temperature is 22°C (Mejri, et al., 2018). (INM, 2014) (ajouter figure temperature pluviometrie annuelle si possible)” (L117-L121).

Comment 6:   The author did not make much contribution to the field of study Please increase resolution of figure 4.

                 Reply 6: We agree with you, in order to mention some limitations of using the two methods “MDC and KDTree-KNM” classifications, the RF showed the best classes distributions, this is consistent with the statistical results obtained by the confusion matrix. However, the two others models, there is some ambiguity between the percentages of class distribution and the land use map obtained. (L261-267).

Comment 7:  This write-up did not address the current limitations in this field of study.

                 Reply 7: Taken into acount your point of view, this current work described the land use map using sentinel 2. Nontheless, we could mention somes limitations in this investigation, because we could incorportate other environmental  data, since this method is very influenced by the environmental parameters. Also, We have already mentioned somes limitations of using RF method in the conclusion section to suggest some perspectives. (L334-337).

Comment 8:   Please add grid to figure 1.

                 Reply 8: As you mentioned, the grid was added to the figure1.  (L130-133)

Comment 9:   In line 256, page 12 using machine learning algorithm machine learning algorithms could be efficient, please remove repeated words.

                   Reply 9Sorry for this mistake, the repeated words “machine learning algorithm” were removed. (L285)

Reviewer 2 Report

I see major challenges in this paper for publication, if the following are well addressed I will give another round of revision.

-       In the paper Land Use and Land Cover (LULC) should be used not LU

-       Last sentences of abstract must be deleted and a proper take home message should be concluded. Sentinel is the only image used in this paper, thus, such conclusion is not realistic

-       Introduction is weak, it should go beyond general statements on the importance of LULC mapping, then describe supervised classification models, why they are important, why they should be compared etc, why you selected these three models and then shed light on your objectives

-       L 59, what is the relevance of this sentence to your work? “According to [20] it is easier 59 to identify floods from optical images than from radar SAR images [22, 21].

-       Study area should much further be extended and describe the main characteristics of the region, population, topography, climate…

-       Data collection should explain the main characteristics of the data in the form of a table, and describe how the data were collected, etc

-       No need to write the bands of Sentiel2, and no need to the table. This section must be merged with the Data collection section

-       Fig.2 must be explained and the main steps should be described briefly

-       Why the image was resampled to 10 m? Why four bands?

-       L147, six classifications > six classes

-       In Supervised Classification Techniques, you have to cite relevant works and explain the main characteristics of the methods in the text not in a table. You can refer to “Google Earth Engine for large-scale land use and land cover mapping: an object-based classification approach using spectral, textural and topographical factors” to get a lot of lessons to describe the RF model and the remaining.  

-       RMSE cannot be used for the accuracy assessment of land use classification. Please use a confusion matrix. All models must be compared, analyzed and discussed using confusion matrix

-       Fig.3 is weak, not clear which map belong to which model

Author Response

First, we would like to thank both reviewers for their constructive and encouraging feedback which improved our manuscript (agriculture-1885143) considerably. We made all the revisions according to reviewer concerns. Please find the answers for each of the points raised by the reviewers in the present MS word file. Please refer to the responses as follows: Reviewers comments in normal font, author’s response in blue, changes in the MS in red.

Reviewer(s)' responses

Reviewer 2:

General comments.

I see major challenges in this paper for publication, if the following are well addressed I will give another round of revision.

Reply: Thank you for all your constitutive remarks. We have fully addressed all following critiques and incorporated the modifications in the revised manuscript.

Comment1:      In the paper Land Use and Land Cover (LULC) should be used not LU.

Reply 1: Sorry for this mistake, the term “LU” was replaced by “LULC”. (L55)

Comment 2: Last sentences of abstract must be deleted and a proper take home message should be concluded. Sentinel is the only image used in this paper, thus, such conclusion is not realistic.

Reply 2: We agree with you, as recommended the last sentence of the abstract was removed and replaced by “Our results suggest that sentinel-2 combined with RF classification is efficient to create LU map. “.(L28-29).

Comment 3: Introduction is weak, it should go beyond general statements on the importance of LULC mapping, then describe supervised classification models, why they are important, why they should be compared etc, why you selected these three models and then shed light on your objectives.

Reply 3: We apologize for this lack of informations, additional data were added to highlight all the importance of LULC mapping. (L56-61, L72-84, L88-104).

We have added some new reference to improve our manuscript:

  1. Abou Samra, R.M. Dynamics of human-induced lakes and their impact on land surface temperature in Toshka Depression, Western Desert, Egypt. Environmental Science and Pollution Research.2022, 29, 20892-20905. [https://doi.org/10.1007/s11356-021-17347-z]

19.Shafizadeh-Moghadam,H,; Khazaei,M,; Alavipanah,S.K,; Weng,Q. Google Earth Engine for large-scale land use and land cover mapping: an object-based classification approach using spectral, textural andtopographical factors. GIScience & Remote Sensing.2021, 58,:914–928 [https://doi.org/10.1080/15481603.2021.1947623].

25.Keuchel,J,; Naumann,S,; Heiler,M,;  Siegmund,A. Automatic land cover analysis for Tenerife by supervised classification using remotely sensed data. Remote Sensing of Environment.2003, 86: 530–541.[ doi:10.1016/S0034-4257(03)00130-5].

  1. Osisanwoer Supervised Machine Learning Algorithms: Classification and Comparison. International Journal of Com-puter Trends and Technology (IJCTT) .2017, 48, Number 3 June 2017:128-138.

        30 Sahebjalal,E,; Dashtekian,K. Analysis of land use-land covers changes using normalized difference vegetation index (NDVI) differencing and classification methods. African Journal of AgriculturalResearch.2013, 8(37): 4614-4622[DOI:10.5897/AJAR11.1825].

  1. Kotisiantis,S.B. Supervised Machine Learning:A review of classification techniques.2007,160:1-24

       34.Talukdar,S, ; Singha,P, ; Mahato,S,; Shahfahad,; Pal,S,; Liou,Y.A,; Rahman,A. Land-Use Land-Cover Classification by Machine Learning Classifiers for SatelliteObservations—A Review. Remote Sens. 2020, 12, 1135 [doi:10.3390/rs12071135).

  1. Kulkarni, A.D, ; Lowe,B. Random Forest Algorithm for Land Cover Classification. International Journal on Recent and Innovation Trends in Computing and Communication.2016,4: 58-63.

      36.Shareef, M. A,; Hassan, N. D,; Hasan, S. F,; Khenchaf, A. Integration of Sentinel-1A and Sentinel-2B Data for Land Use and Land Cover Mapping of the Kirkuk Governorate, Iraq. International Journal of Geoinformatics.2020,16 :87-96.

      37.Ameur,M,; Hamzaoui-Azaza,F, ; Gueddari,M. Suitability for human consumption and agriculture purposes of Sminja aqui-fer groundwater in Zaghouan (north-east of Tunisia) using GIS and geochemistry techniques. Environ Geochem Health .2016,38:1147–1167.[ DOI 10.1007/s10653-015-9780-2].

  1. Mejri,S,; Chekirbene, A,; Tsujimura ,M,; Boughdiri, M,; Mlayah, A. Tracing groundwater salinization processes in an inland aquifer: A hydrogeochemical and isotopic approach in Sminja aquifer (Zaghouan, northeast of Tunisia). Journal of African Earth Sciences.2017:1-39. [10.1016/j.jafrearsci.2018.07.009].

Comment 4: L 59, what is the relevance of this sentence to your work? “According to [20] it is easier 59 to identify floods from optical images than from radar SAR images [22, 21].”

Reply 4: Sorry for this mistake and as mentioned in the response to the fisrt reviewer, the sentence “According to [20] it is easier 59 to identify floods from optical images than from radar SAR images [22, 21]” was removed.

Comment 5: Study area should much further be extended and describe the main characteristics of the region, population, topography, climate…

Reply 5: As suggested, more informations were added to describe the main characteristics of the region as “climate, vegetation, geology, topography, evapotraspiration”. (L115-125).

Comment 6:  Data collection should explain the main characteristics of the data in the form of a table, and describe how the data were collected, etc.

Reply 6: As mentioned, the main characteristics of the data were added in the form of table describing more details about data collections.(L150-153)

Comment 7: No need to write the bands of Sentiel2, and no need to the table. This section must be merged with the Data collection section.

Reply 7: Based in your comment, we incorportate the bands of Sentiel-2 and the table in the data collection section. (L143-147)

Comment 8: Fig.2 must be explained and the main steps should be described briefly.

Reply 8: As requested, Figure 2 was already well clarified in the text, particularly in the methodology part (2.4.1-2.4.3) and the main steps were briefly described as recommend.

Comment 9: Why the image was resampled to 10 m? Why four bands?

Reply 9: Sorry for this omission, we have five bands. We resampled to 10 m because the indices we used need in their equations the bands with a resolution of 10m. In addition resampled to 10m as chosen in order to improve the resolution. (L172-173).

Comment 10: L147, six classifications > six classes.

Reply 10: We changed “six classifications bysix classes”.  (L169)

     Comment 11:  In Supervised Classification Techniques, you have to cite relevant works and explain the main characteristics of the methods in the text not in a table. You can refer to “Google Earth Engine for large-scale land use and land cover mapping: an object-based classification approach using spectral, textural and topographical factors” to get a lot of lessons to describe the RF model and the remaining.  

Reply 11: Based on your comments, some modifications were added, relevant works were mentioned in the text. The main characteristic were explained in the text. (L191-211).

Comment 12: RMSE cannot be used for the accuracy assessment of land use classification. Please use a confusion matrix. All models must be compared, analyzed and discussed using confusion matrix.

Reply 12:  As suggested, the appropriate changes was added, we already used a  confusion matrix but it was not well explained, thus some modifications were added to clarify this sub-section. (L214)

Comment 13: Fig.3 is weak, not clear which map belong to which model.

Reply 13: I m agree with you, because of the lack of some informations in the fig 3, we tried to more explain it. Thus, references were addedd in each land use map. (L257-260)

Round 2

Reviewer 1 Report

I recommend accepting the manuscript in the present form

Reviewer 2 Report

The manuscript has been improved significantly.